# The neural dynamics of auditory word recognition and integration

**Jon Gauthier** and **Roger Levy**
Department of Brain and Cognitive Sciences
Massachusetts Institute of Technology
jon@gauthiers.net, rplevy@mit.edu

## Abstract

Listeners recognize and integrate words in rapid and noisy everyday speech by combining expectations about upcoming content with incremental sensory evidence. We present a computational model of word recognition which formalizes this perceptual process in Bayesian decision theory. We fit this model to explain scalp EEG signals recorded as subjects passively listened to a fictional story, revealing both the dynamics of the online auditory word recognition process and the neural correlates of the recognition and integration of words.

The model reveals distinct neural processing of words depending on whether or not they can be quickly recognized. While all words trigger a neural response characteristic of probabilistic integration — voltage modulations predicted by a word's surprisal in context — these modulations are amplified for words which require more than roughly 150 ms of input to be recognized. We observe no difference in the latency of these neural responses according to words' recognition times. Our results are consistent with a two-part model of speech comprehension, combining an eager and rapid process of word recognition with a temporally independent process of word integration. However, we also developed alternative models of the scalp EEG signal not incorporating word recognition dynamics which showed similar performance improvements. We discuss potential future modeling steps which may help to separate these hypotheses.

Psycholinguistic studies at the neural and behavioral levels have detailed how listeners actively predict upcoming content at many levels of linguistic representation (Kuperberg and Jaeger, 2016), and use these predictions to drive their behavior far before the relevant linguistic input is complete (Allopenna et al., 1998). One well-studied neural correlate of this prediction-driven

comprehension process is the N400 ERP, a centro-parietally distributed negative voltage modulation measured at the scalp by electroencephalogram (EEG) which peaks around 400 ms after the onset of a word. This negative component is amplified for words which are semantically incompatible with their sentence or discourse context (Kutas and Hillyard, 1984; Brown and Hagoort, 1993; Kutas and Federmeier, 2011; Heilbron et al., 2022). This effect has been taken as evidence that comprehenders actively predict features of upcoming words (DeLong et al., 2005; Kuperberg and Jaeger, 2016; Kuperberg et al., 2020). On one popular account, predictions about upcoming content are used to pre-activate linguistic representations likely to be used when that content arrives. The N400 reflects the integration of a recognized word with its context, and this integration is facilitated just when the computational paths taken by the integration process align with those already pre-activated by the listener (Kutas and Federmeier, 2011; Federmeier, 2007).

Despite the extensive research on the N400 and its computational interpretation, its relationship with the upstream process of word recognition is still not well understood. Some authors have argued that integration processes should be temporally yoked to word recognition: that is, comprehenders should continue gathering acoustic evidence as to the identity of a word until they are sufficiently confident to proceed with subsequent integration processes (Marslen-Wilson, 1987). It is also possible, however, that integration processes are insensitive to the progress of word recognition: that integration is a temporally regular semantic operation which begins regardless of the listener's confidence about the word being spoken (Hagoort, 2008; Federmeier and Laszlo, 2009).

Experimental studies have attempted to assess the link between these two processes, modeling the timing of word recognition through an offline

---

Code to reproduce our analyses is available at github.com/hans/word-recognition-and-integration.

behavioral paradigm known as *gating* (Grosjean, 1980): by presenting incrementally longer clips of speech to subjects and asking them to predict what word is being spoken, authors estimate the time point at which there is sufficient information to identify a word from its acoustic form. Several EEG studies have asked whether the N400 response varies with respect to this estimate of word recognition time, but have arrived at contradictory answers to this question (van den Brink et al., 2006; O'Rourke and Holcomb, 2002).

In this paper, we introduce a computational model which targets these dynamics of word recognition, and their manifestation in neural EEG signals recorded during naturalistic listening. The model allows us to connect trial-level variation in word recognition times to aspects of the neural response to words. We use the model to address two cross-cutting questions:

- **Onset:** Are words integrated only after they are successfully recognized, or is the timing of integration insensitive to the state of word recognition?

- **Response properties:** Does the shape of the neural response to words differ based on their recognition times? If so, this could indicate distinct inferential mechanisms deployed for words depending on their ease of recognition.

We jointly optimize the cognitive and neural parameters of this model to explain EEG data recorded as subjects listened to naturalistic English speech. Model comparison results suggest that semantic integration processes are not temporally yoked to the status of word recognition: the neural traces of word integration have just the same temporal structure, regardless of when words are successfully recognized. However, the neural correlates of word integration qualitatively differ based on the status of word recognition: words not yet recognized by the onset of word integration exhibit significantly different neural responses.

These results suggest a two-part model of word recognition and integration. First, the success of our word recognition model in predicting the neural response to words suggests that there exists a rapid lexical interpretation process which integrates prior expectations and acoustic evidence in order to pre-activate specific lexical items in memory. Second, an independent integration process composes these memory contents with a model of

| | Meaning | Bounds |
|---|---|---|
| $\gamma$ | Recognition threshold (eq. 3) | $(0,1)$ |
| $\lambda$ | Evidence temperature (eq. 2) | $(0,\infty)$ |
| $\alpha$ | Scatter point (eq. 4) | $(0,1)$ |
| $\alpha_p$ | Prior scatter point (eq. 4) | $(0,1)$ |
| $k_i^*$ | Word $w_i$'s recognition point (eq. 3) | $\{0,1,\ldots,|w_i|\}$ |
| $\tau_i$ | Word $w_i$'s recognition time (eq. 4) | $[0,\infty)$ |

Table 1: Cognitive model parameters and outputs.

the context, following a clock which is *insensitive* to the specific state of word recognition.

It is necessary to moderate these conclusions, however: we also develop alternative models of the neural correlates of word integration which improve beyond the performance of our baselines, without incorporating facts about the dynamics of word recognition. We discuss in Section 4 how more elaborate neural linking theories will be necessary to better separate these very different cognitive pictures of the process of word recognition and its neural correlates.

## 1 Model

Our model consists of two interdependent parts: a cognitive model of the dynamics of word recognition, and a neural model that estimates how these dynamics drive the EEG response to words.

### 1.1 Cognitive model

We first design a cognitive model of the dynamics of word recognition in context, capturing how a listener forms incremental beliefs about the word they are hearing $w_i$ as a function of the linguistic context $C$ and some partial acoustic evidence $I_{\leq k}$. We formalize this as a Bayesian posterior (Norris and McQueen, 2008):

$$P(w_i \mid C, I_{\leq k}) \propto P(w_i \mid C)\, P(I_{\leq k} \mid w_i) \quad (1)$$

which factorizes into a prior expectation of the word $w_i$ in context (first term) and a likelihood of the partial evidence of $k$ phonemes $I_{\leq k}$ (second term). This model thus asserts that the context $C$ and the acoustic input $I_{\leq k}$ are conditionally independent given $w_i$. We parameterize the prior $P(w_i \mid C) = P(w_i \mid w_{<i})$ using a left-to-right neural network language model. The likelihood is a noisy-channel phoneme recognition model:

$$P(I_{\leq k} \mid w_i) \propto \prod_{1 \leq j \leq k} P(I_j \mid w_{ij})^{\frac{1}{\lambda}} \quad (2)$$

where per-phoneme confusion probabilities are drawn from prior phoneme recognition studies (Weber and Smits, 2003) and reweighted by a temperature parameter $\lambda$.

We evaluate this posterior for every word with each incremental phoneme, from $k = 0$ (no input) to $k = |w_i|$ (conditioning on all of the word's phonemes). We define a hypothetical cognitive event of *word recognition* which is time-locked to the phoneme $k_i^*$ where this posterior first exceeds a confidence threshold $\gamma$:

$$k_i^* = \min_{0 \leq k \leq |w_i|} \{k \mid P(w_i \mid C, I_{\leq k}) > \gamma\} \quad (3)$$

We define a word's *recognition time* $\tau_i$ to be a fraction $\alpha$ of the span of the $k_i^*$-ith phoneme. In the special case where $k_i^* = 0$ and the word is confidently identified prior to acoustic input, we take $\tau_i$ to be a fraction $\alpha_p$ of its first phoneme's duration (visualized in Figure 1a):

$$\tau_i = \begin{cases} \text{ons}_i(k_i^*) + \alpha \, \text{dur}_i(k_i^*) & \text{if } k_i^* > 0 \\ \alpha_p \, \text{dur}_i(1) & \text{if } k_i^* = 0 \end{cases} \quad (4)$$

where $\text{ons}_i(k)$ and $\text{dur}_i(k)$ are the onset time (relative to word onset) and duration of the $k$-th phoneme of word $i$, and $\alpha, \alpha_p$ are free parameters fitted jointly with the rest of the model.

## 1.2 Neural model

We next define a set of candidate linking models which describe how the dynamics of the cognitive model (specifically, word recognition times $\tau_i$) affect observed neural responses. These models are all variants of a temporal receptive field model (TRF; Lalor et al., 2009; Crosse et al., 2016), which predicts scalp EEG data over $S$ sensors and $T$ samples, $Y \in \mathbb{R}^{S \times T}$, as a convolved set of linear responses to lagged features of the stimulus:

$$Y_{st} = \sum_f \sum_{\Delta=0}^{\tau_f} \Theta_{f,s,\Delta} \times \mathbf{X}_{f,t-\Delta} + \epsilon_{st} \quad (5)$$

where $\tau_f$ is the maximum expected lag (in seconds) between the onset of a feature $f$ and its correlates in the neural signal; and the inner sum is accumulated in steps of the relevant neural sampling rate. This deconvolutional model estimates a characteristic linear response linking each feature of the stimulus to the neural data over time. The model allows us to effectively uncover the neural response to individual stimulus features in naturalistic data, where stimuli (words) arrive at a fast

| Model name | Onset | Response properties |
|---|---|---|
| Baseline | 0 | unitary linear response |
| Shift | $\tau_i$ (eq. 4) | unitary linear response |
| Variable | 0 | independent linear responses for early-, mid-, and late-recognized words |
| Prior-variable | 0 | independent linear responses for low-, mid-, and high-surprisal words |

Table 2: Neural linking models with different commitments about the temporal onset of word features (relative to word onset) and the flexibility of the parameters linking word features to neural response.

rate, and their neural responses are likely highly convolved as a consequence (Crosse et al., 2016).

We define a feature time series $X_t \in \mathbb{R}^{d_t \times T}$ containing $d_t$ features of the objective auditory stimulus, such as acoustic and spectral features, resampled to match the $T$ samples of the neural time series. We also define a word-level feature matrix $X_v \in \mathbb{R}^{d_w \times n_w}$ for the $n_w$ words in the stimulus. Crucially, $X_v$ contains estimates of each word's surprisal (negative log-probability) in context. Prior studies suggest that surprisal indexes the peak amplitude of the naturalistic N400 (Frank et al., 2015; Gillis et al., 2021; Heilbron et al., 2022).

We assume that $X_t$ causes a neural response independent of word recognition dynamics, while the neural response to features $X_v$ may vary as a function of recognition dynamics. These two feature matrices will be merged together to yield the design matrix $\mathbf{X}$ in Equation 5.

We enumerate several possible classes of neural models which describe different ways that a word's recognition time $\tau_i$ may affect the neural response. Each model class constitutes a different answer to our framing questions of **onset** and **response properties** (Table 2 and Figure 1b), by specifying different featurizations of word-level properties $X_v$ in the TRF design matrix $\mathbf{X}$:

1. Unitary response aligned to word onset (*baseline model*): All words exhibit a unitary linear neural response to recognition and integration, time-locked to the word's onset in the stimulus. This baseline model, which does not incorporate the cognitive dynamics of recognition in any way, is what has been assumed by prior naturalistic modeling work.

   This model asserts that each word's features

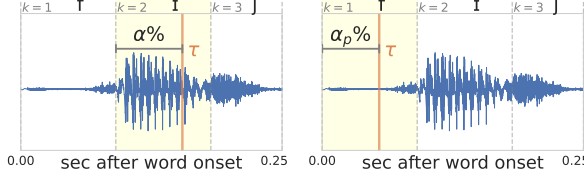

(a) Computation of recognition time $\tau_i$ for a recognition point after phoneme $k_i^* = 2$ (left) or recognition prior to input, $k_i^* = 0$ (right) for a spoken word *fish* /fɪʃ/. See eq. 4.

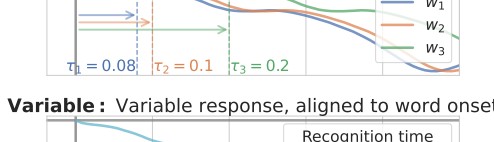

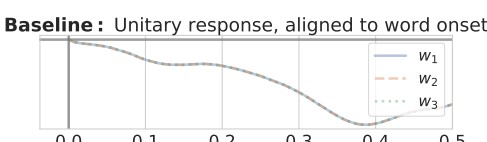

(b) Candidate neural model logic linking three words' recognition times $\tau_i$ to neural modulations by surprisal.

Figure 1: Sketches of model logic.

$X_{vi}$ trigger a neural response beginning at the onset of word $i$, and that this neural response can be captured by a single characteristic response to all words.

2. Unitary response aligned to recognition time (*shift model*): All words exhibit a unitary linear neural response to recognition and integration, time-locked to the word's recognition time $\tau_i$.

   This model asserts that each word's features $X_{vi}$ trigger a neural response beginning at $\tau_i$ seconds after word onset, and that this neural response can be captured by a single characteristic response to all words.

3. Variable response by recognition time, aligned to word onset (*variable model*): Words exhibit a differential neural response to recognition and integration based on their recognition time. The temporal onset of these integration processes is insensitive to the progress of word recognition.

   We account for variable responses by defining a quantile split $Q : \tau \to \mathbb{N}$ on the inferred recognition times $\tau_i$. We then estimate distinct TRF parameters for the features of

words in each quantile.

This model thus asserts that it is possible to group words by their recognition dynamics such that they have a characteristic neural response within-group, but differ freely between groups.

4. Variable response by word surprisal, aligned to word onset (*prior-variable model*): This model is identical to the above *variable model*, except that words are divided into quantiles based on their surprisal in context rather than their recognition time.

   This model instantiates the hypothesis that the shape of the neural response to words varies based on listeners' expectations, but only those driven by the preceding linguistic context. On this reading, words are preactivated according to their *prior* probability, rather than their rapidly changing *posterior* probability under some acoustic input.[1]

For a set of recognition time predictions $\tau_i$, we estimate within-subject TRFs under each of these linking models, yielding per-subject parameters $\Theta_j$, describing the combined neural response to objective stimulus features and word-level features. This estimation procedure allows for within-subject variation in the shape of the neural response.

## 2 Methods and dataset

We jointly infer[2] across-subject parameters of the cognitive model (Table 1) and within-subject parameters of the neural model in order to minimize regularized L2 loss on EEG data, estimated by 4-fold cross-validation. We then compare the fit models on held-out test data, containing 25% of the neural time series data for each subject. For each comparison of models $m_1, m_2$, we compute the Pearson correlation coefficient $r$ between the predicted and observed neural response for each subject at each EEG sensor $s$. We then use paired $t$-tests to ask whether the within-subject difference

---

[1]This reading is compatible with pre-activation theories (e.g. Brothers and Kuperberg, 2021). At their present level of specificity, it is unclear whether this focus on prior probability is a substantive commitment, or simply a choice of modeling expediency.

[2]We conduct tree-structured Parzen estimator random search (Bergstra et al., 2011) with Optuna (Akiba et al., 2019).

in $r$ pooled across sensors significantly differs between $m_1$ and $m_2$:

$$\frac{1}{S}\sum_{s=1}^{S} r\left(Y_s, \hat{Y}_{m_1,s}\right) \overset{?}{>} \frac{1}{S}\sum_{s=1}^{S} r\left(Y_s, \hat{Y}_{m_2,s}\right) \quad (6)$$

**Dataset**  We analyze EEG data recorded as 19 subjects listened to Hemingway's *The Old Man and the Sea*, published in Heilbron et al. (2022). The 19 subjects each listened to the first hour of the recorded story while maintaining fixation. We analyze 5 sensors distributed across the centro-parietal scalp: one midline sensor and two lateral sensors per hemisphere at central and posterior positions. The EEG data were acquired using a 128-channel ActiveTwo system at a rate of 512 Hz, and down-sampled offline to 128 Hz and re-referenced to the mastoid channels. We follow the authors' preprocessing method, which includes band-pass filtering the EEG signal between 0.5 and 8 Hz, visual annotation of bad channels, and removal of eyeblink components via independent component analysis.[3] The dataset also includes force-aligned annotations for the onsets and durations of both words and phonemes in these time series.

We generate a predictor time series $X_t$ aligned with this EEG time series (Appendix B), ranging from stimulus features (features of the speech envelope and spectrogram) to sublexical cognitive features (surprisal and entropy over phonemes). By including these control features in our models, we can better understand whether or not there is a cognitive and neural response to words distinct from responses to their constituent properties (see Section 4.2 for further discussion). We generate in addition a set of word-level feature vectors $X_v \in \mathbb{R}^{3 \times n_w}$, consisting of an onset feature and

1. word surprisal in context, computed with GPT Neo 2.7B (Black et al., 2021),[4] and
2. word unigram log-frequency, from SUB-TLEXus 2 (Brysbaert and New, 2009).

**Likelihood estimation**  Our cognitive model requires an estimate of the confusability between English phonemes (Equation 2). We draw on the experimental data of Weber and Smits (2003),

---

[3]See Appendix E for further details on our choice of band-pass filter width.

[4]Preliminary experiments using our baseline model showed that surprisal estimates from GPT Neo 2.7B best explained held-out EEG signals, compared among other sizes of GPT Neo and OpenAI GPT-2 models (Radford et al., 2019; Brown et al., 2020).

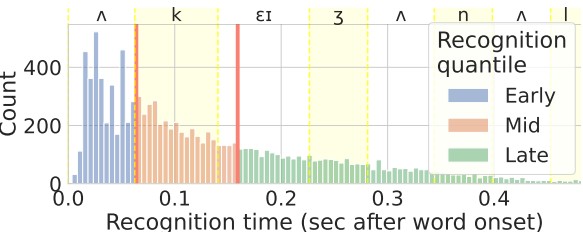

Figure 2: Distribution of inferred recognition times (relative to word onset) for all words, as predicted by the optimal cognitive model parameters. Salmon vertical lines indicate a tertile partition of words by their recognition time; light yellow regions indicate the median duration of phonemes at each integer position within a word. An example stimulus word, *occasional*, is aligned with phoneme duration regions above the graph.

who estimated patterns of confusion in phoneme recognition within English consonants and vowels by asking subjects to transcribe spoken syllables. Their raw data consists of count matrices $\psi_c, \psi_v$ for consonants and vowels, respectively, where each cell $\psi[ij]$ denotes the number of times an experimental subject transcribed phoneme $j$ as phoneme $i$, summing over different phonological contexts (syllable-initial or -final) and different levels of acoustic noise in the stimulus presentation. We concatenate this confusion data into a single matrix, imputing a count of 1 for unobserved confusion pairs, and normalize each column to yield the required conditional probability distributions.

## 3  Results

We first evaluate the baseline model relative to a TRF model which incorporates no word-level features $X_v$ except for a word onset feature, and find that this model significantly improves in held-out prediction performance ($t = 4.91, p = 0.000113$). The model recovers a negative response to word surprisal centered around 400 ms post word onset (Figure 6), which aligns with recent EEG studies of naturalistic language comprehension in both listening (Heilbron et al., 2022; Gillis et al., 2021; Donhauser and Baillet, 2020) and reading (Frank et al., 2015).

We next separately infer optimal model parameters for the shift and variable models, and evaluate their error on held-out test data. We find that the variable model significantly exceeds the baseline model ($t = 5.15, p = 6.70 \times 10^{-5}$), while the shift

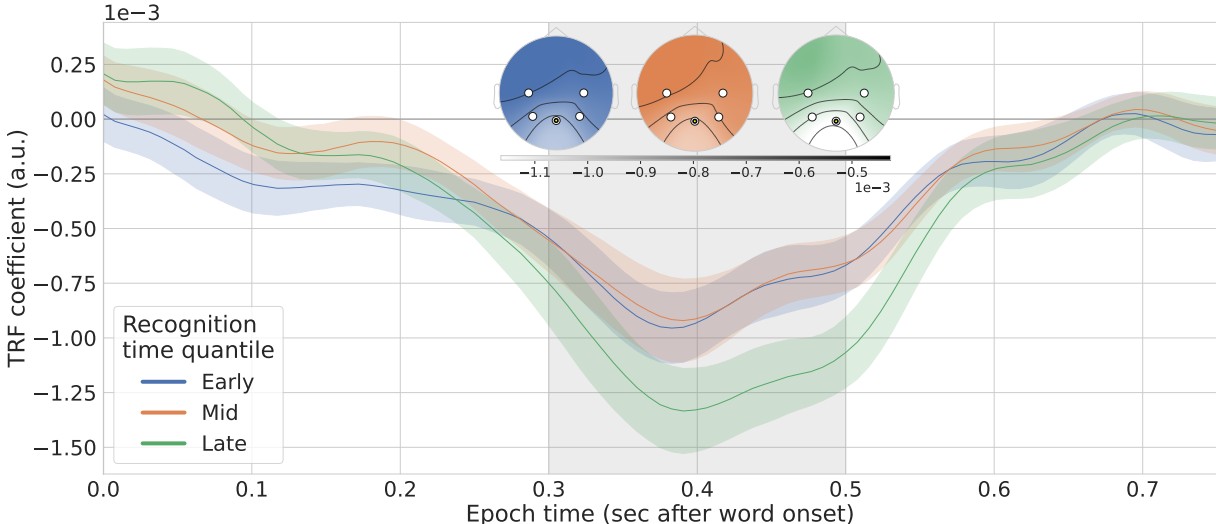

Figure 3: Modulation of scalp voltage at a centro-parietal site by surprisal for words with early ($< 64$ ms, blue), middle ($< 159$ ms, orange), or late ($> 159$ ms, green) recognition times. Lines denote inferred coefficients of word surprisal in averaged over subjects for the sensor highlighted in the inset. Error regions denote s.e.m. ($n = 19$). Inset: spatial distribution of surprisal modulations averaged for each recognition time quantile within vertical gray regions, where less saturated colors denote more negative response. The surprisal response peaks ∼400 ms post onset, amplified for late-recognized words (green).

model does not ($t = 2.23, p = 0.039$).[5] This suggests that neural responses to words are not simply temporally yoked to their recognition times.

We next investigate the parameters of the optimal variable model. Figure 2 shows the distribution of predicted word recognition times $\tau_i$ under the optimal variable model on stimulus data from the held-out test set, charted relative to the onset of a word. Our model predicts that one third of words are recognized prior to 64 ms post word onset, another third are recognized between 64 ms and 159 ms, and a long tail are recognized after 159 ms post word onset. This entails that at least a third of words are recognized prior to any meaningful processing of acoustic input. This prediction aligns with prior work in multiple neuroimaging modalities, which suggests that listeners pre-activate features of lexical items far prior to their acoustic onset in the stimulus (Wang et al., 2018; Goldstein et al., 2022).

These inferred recognition times maximize the likelihood of the neural data under the linking variable model parameters $\Theta$. Figure 3 shows the variable model's parameters describing a neural response to word surprisal for each of three recognition time quantiles, time locked to word onset. We see two notable trends in the N400 response

which differ as a function of recognition time:

1. Figure 3 shows word surprisal modulations estimated at a centro-parietal site for the three recognition time quantiles. Words recognized late (159 ms or later post word onset) show an exaggerated modulation due to word surprisal. The peak negative amplitude of this response is significantly more negative than the peak negative response to early words (fig. 3, green line peak minus blue line peak in the shaded region; within-subject paired $t = -5.23, p = 5.71 \times 10^{-5}$). This modulation is spatially distributed similarly to the modulation for early-recognized words (compare the green inset scalp distribution to that of the blue and orange scalps).

2. There is no significant difference in the latency of the N400 response for words recognized early vs. late. The time at which the surprisal modulation peaks negatively does not significantly differ between early and late words (fig. 3, green line peak time minus blue line peak time; within-subject paired $t = 2.17, p = 0.0440$).

These model comparisons and analyses of optimal parameters yield answers to our original questions about the dynamics of word recognition and integration:

---

[5]A direct comparison of the variable model and shift model performance also favors the variable model ($t = 5.49, p = 3.24 \times 10^{-5}$).

**Response properties:** Neural modulations due to surprisal are exaggerated for words recognized late after their acoustic onset.

**Onset:** The variable model, which asserted integration processes are initiated relative to words' onsets rather than their recognition times, demonstrated a better fit to the data. The optimal parameters under the variable model further showed that while word recognition times seem to affect the amplitude of neural modulations due to surprisal, they do not affect their latency.

### 3.1 Prior-variable model

We compute a surprisal-based quantile split over words in the training dataset. The first third of low-surprisal words had a surprisal lower than 1.33 bits, while the last third of high-surprisal words had a surprisal greater than 3.71 bits.

We next estimate the prior-variable neural model parameters, which describe independent neural responses to words in low-, mid-, and high-surprisal quantiles. This model also significantly exceeds the baseline model ($t = 7.78, p = 3.64 \times 10^{-7}$; see Appendix C for inferred model parameters). Figure 4 shows a comparison of the way the prior-variable model and the variable model sorted words into different quantiles. While the two models rarely made predictions at the opposite extremes (labeling a low-surprisal word as late-recognized, or a high-surprisal word as early-recognized; bottom left and upper right black corners in fig. 4a), there were many disagreements involving sorting words into neighboring time bins (off-diagonal in fig. 4a). Figures 4b and 4c show some meaningful cases in which the models disagree to be due to differences in the relevant phonological neighborhood early in the onset of a word. Figure 4c shows the recognition model's posterior belief over words (eq. 1) given the incremental phonetic input at the top of the graph. The left panel of Figure 4c shows how the word *disgust* is recognized relatively late due to a large number of contextually probable phonological neighbors (such as *dismay* and *despair*); the right panel shows how the word *knelt* is recognizable relatively early, since most of the contextually probable completions (*took*, *had*) are likely to be ruled out after the presentation of a second phone.

The variable model's generalization performance is not significantly different than that of this prior-variable model ($t = -0.422, p = 0.678$).

Future work will need to leverage other types of neural data to distinguish these models. We discuss this further in Section 4 and the Limitations section.

## 4 Discussion

This paper presented a cognitive model of word recognition which yielded predictions about the recognition time of words in context $\tau_i$. A second neural linking model, the variable model, estimated the neural response to words recognized at early, intermediate, and late times according to the cognitive model's predictions. This latter model significantly improved in held-out generalization performance over a baseline model which did not allow for differences in the neural signal as a function of a word's recognition time. We also found, however, that a neural model which estimated distinct shapes of the neural response to words based on their surprisal — not their recognition times — also improved beyond our baseline, and was indistinguishable from the variable model. More elaborate neural linking theories describing how words' features drive the neural response will be necessary to distinguish these models (see e.g. the encoding model of Goldstein et al., 2022).

Our positive findings are consistent with a two-part model of auditory word recognition and integration, along the lines suggested by van den Brink et al. (2006) and Hagoort (2008, §3c). In this model, listeners continuously combine their expectations with evidence from sensory input in order to load possible lexical interpretations of the current acoustic input into a memory buffer. Our model's prediction of a word's recognition time $\tau_i$ measures the time at which this buffer resolves in a clear lexical inference.

A second integration process reads out the contents of this buffer and merges them with representations of the linguistic context. Our latency results show that the timing of this process is independent of a listener's current confidence in their lexical interpretations, instead time-locked to word onset. This integration process thus exhibits two distinct modes depending on the listener's buffer contents: one *standard*, in which the buffer is clearly resolved, and one *exceptional*, in which the buffer contents are still ambiguous, and additional inferential or recovery processes must be deployed in order to proceed with integration. Future work could spell out this distinction mech-

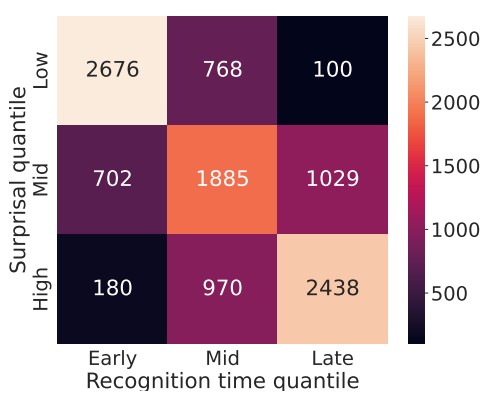

| Context | Prior-only prediction | Rec. time prediction |
|---|---|---|
| …he looked at it in *disgust* | Mid | Late |
| …the old man was now definitely and *finally* | Mid | Late |
| …drew his knife across one of the *strips* | Mid | Late |
| …on his cheeks. The *blotches* | High | Mid |
| ¶He *knelt* | High | Mid |
| ¶"*I* | Mid | Early |

(b) Examples of disagreements in word labeling between the prior-only model and the recognition model.

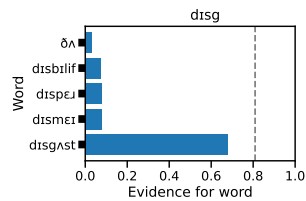 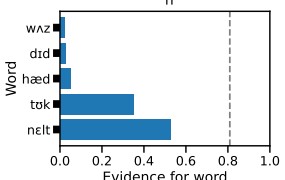

(a) Confusion matrix comparing partitions of words by the prior-variable model (based on word surprisal; vertical axis) and the optimal word recognition model (based on recognition time; horizontal axis).

(c) Example posterior predictive distributions for words recognized late due to a dense neighborhood (left); and early due to a sparse neighborhood (right).

Figure 4: Differing predictions of the word recognition model and the prior-variable (surprisal-based) model.

anistically in order to explain how buffers in the "exceptional" state elicit these distinct neural responses.

## 4.1 What determines integration timing?

Our findings on the stable timing of the naturalistic N400 align with some prior claims in the experimental ERP literature (Federmeier and Laszlo, 2009, §5).[6] These results strengthen the notion that, even in rapid naturalistic environments, the timing of the early semantic integration of word meanings is driven not by when words are recognized, but rather by the tick of an external clock.

If this integration process is not sensitive to the status of word recognition, then what drives its dynamics? Federmeier and Laszlo (2009) argue that this regularly timed integration process is language-external, functioning to bind early representations of word meaning with existing cognitive representations of the context via temporal synchrony (see also Kutas and Federmeier, 2011). However, other language-internal mechanisms are also compatible with the data. Listeners may adapt to low-level features of the stimulus, such as their counterpart's speech rate or prosodic cues, manipulating the timing of integration to maximize the

chances of success in the expected case.[7]

Alternatively, listeners may use the results of the word recognition process to schedule upcoming attempts at word integration. After recognizing each word $w_i$, listeners may form an expectation about the likely onset time of word $w_{i+1}$, using knowledge about the form of $w_i$ and the speech rate. Listeners could instantiate a clock based on this prediction, counting down to a time some fixed distance from the expected onset of $w_{i+1}$, at which semantic integration would be most likely to succeed on average. Such a theory could explain how word recognition and integration are at least approximately optimal given limited cognitive resources (Simon, 1955; Lieder and Griffiths, 2020): they are designed to successfully process linguistic inputs in expectation, under the architectural constraint of a fixed integration clock.

## 4.2 Words as privileged units of processing

Our results suggest that words exist at a privileged level of representation and prediction during speech processing. This is not a necessary property of language processing: it is possible that word-level processing effects (neural or behavioral responses to word-level surprisal) could emerge as an epiphenomenon of *lower-level* pre-

---

[6]This is a claim about the *within-subject* consistency of N400 timing, despite substantial between-subject variability, for example, by age and language experience (Federmeier and Laszlo, 2009).

[7]See Verschueren et al. (2022, Figure 6 and Table 4) for evidence against this point, demonstrating that controlled variation in stimulus speech rate does not affect the latency of the N400 response.

diction and integration of sublexical units, e.g., graphemes or phonemes. Smith and Levy (2013, §2.4) illustrate how a "highly incremental" model which is designed to predict and integrate sublexical units (grapheme- or phoneme-based prediction) but which is measured at higher levels (in word-level reading times or word-level neural responses) could yield apparent contrasts that are suggestive of word-level prediction and integration. On this argument, neural responses to word-level surprisal are not alone decisive evidence for word-level prediction and integration (versus the prediction and integration of sub-lexical units).

Our results add a critical orthogonal piece of evidence in favor of word-level integration: we characterized an integration architecture whose timing is locked to the appearance of word units in the stimulus. While the present results cannot identify the precise control mechanism at play here (section 4.1), the mere fact that words are the target of this timing process indicates an architecture strongly biased toward word-level processing.

### 4.3  Prospects for cognitive modeling

The cognitive model of word recognition introduced in this paper is an extension of Shortlist B (Norris and McQueen, 2008), a race architecture specifying the dynamics of single-word recognition within sentence contexts. We used neural network language models to scale this model to describe naturalistic speech comprehension. While we focus here on explaining the neural response to words, future work could test the predictions of this model in behavioral measures of the dynamics of word recognition, such as lexical decision tasks (Tucker et al., 2018; Ten Bosch et al., 2022).

## 5  Conclusion

This paper presented a model of the cognitive and neural dynamics of word recognition and integration. The model recovered the classic N400 integration response, while also detecting a distinct treatment of words based on how and when they are recognized: words not recognized until more than 150 ms after their acoustic onset exhibit significantly amplified neural modulations by surprisal. Despite this processing difference, we found no distinction in the latency of integration depending on a word's recognition time.

However, we developed an alternative model of the neural signal not incorporating word recog-

nition dynamics which also exceeded baseline models describing the N400 integration response. More substantial linking hypotheses bridging between the cognitive state of the word recognition model and the neural signal will be necessary to separate these distinct models.

## Limitations

There are several important methodological limitations to the analyses in this paper.

We assume for the sake of modeling expediency that all listeners experience the same word recognition dynamics in response to a linguistic stimulus. Individual differences in contextual expectations, attention, and language knowledge certainly modulate this process, and these differences should be accounted for in an elaborated model.

We also assume a relatively low-dimensional neural response to words, principally asserting that the contextual surprisal of a word drives the neural response. This contrasts with other recent brain mapping evaluations which find that high-dimensional word representations also explain brain activation during language comprehension (Goldstein et al., 2022; Caucheteux and King, 2022; Schrimpf et al., 2021). A more elaborate neural linking model integrating higher-dimensional word representations would likely allow us to capture much more granular detail at the cognitive level, describing how mental representations of words are retrieved and integrated in real time. Such detail may also allow us to separate the two models (the variable and prior-variable models) which were not empirically distinguished by the results of this paper.

## Acknowledgments

We thank Aixiu An, Jacob Andreas, Canaan Breiss, Trevor Brothers, Tyler Brooke Wilson, Samer Nour Eddine, Evelina Fedorenko, Micha Heilbron, Shailee Jain, Peng Qian, Cory Shain, Jakub Szewczyk, and Josh Tenenbaum for comments on earlier versions of this paper. We thank Micha Heilbron, Marlies Gillis, and Tamar Regev for invaluable advice on EEG data analysis, and for sharing analysis code and data. JG gratefully acknowledges support from the Open Philanthropy Project and RPL gratefully acknowledges support from a Newton Brain Science Research Seed Award.

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

## A  Relation to pre-activation accounts

Our theoretical account discussed in Section 4 is partly compatible with *pre-activation* accounts of prediction in language comprehension, which likewise suggest that listeners eagerly pre-activate features at multiple levels of linguistic representation, according to both contextual expectations and partial sensory input (see e.g. Federmeier (2007); Federmeier and Laszlo (2009); Kutas and Federmeier (2011); Kuperberg and Jaeger (2016) for reviews). Our cognitive model of word recognition provides a mechanism for the temporal dynamics of this pre-activation process. This mechanism is an aggressively incremental process, depending on a probabilistic inference which repeatedly integrates novel acoustic evidence with existing expectations drawn from the context.

Pre-activation accounts suggest that what is pre-activated are abstract semantic features rather than specific lexical items (Federmeier and Kutas, 1999; Kuperberg and Jaeger, 2016). The present

model is stated at the computational level and is thus not directly comparable in this respect. Future modeling work can instantiate specific representational alternatives within this predictive word recognition model and explore how their predictions might settle these questions.

## B  Model featurization

We use a subset of the sublexical features from Heilbron et al. (2022) in our TRF models (named as $X_t$ in Section 1.2). These features are shared across all models tested in our main and baseline analysis:

- onset features for each phoneme in the audio stimulus;

- phoneme-onset aligned features:
  - acoustic control features, averaged within the span of a phoneme: average variance in the broadband envelope, and spectral power measures averaged within eight bins spaced evenly on a log-mel scale
  - the entropy over a next-phoneme distribution $P(p_j \mid w_{i,<j})$ and the surprisal of the ground-truth phoneme, using the hierarchical predictive model of Heilbron et al. (2022) (see below).

### B.1  Phoneme probability estimator

The phoneme model of Heilbron et al. (2022), whose surprisal and entropy measures we use as control predictors, combines a word-level language model prior and a cohort-based likelihood. For some prior phoneme sequence $p_1, \ldots, p_{t-1}$ and some incoming phoneme $p_t$ in a linguistic context $C$, we define

$$
\begin{aligned}
&P(p_t \mid p_1, \ldots, p_{t-1}, C) \\
&\propto \sum_{w \in V} P(w \mid C, p_1, \ldots, p_{t-1}) \, P(p_t \mid w) \\
&= \sum_{w \in V} P(w \mid C) \, \mathbf{1}\{w \in \mathrm{Coh}(p_1, \ldots, p_{t-1}, p_t)\}
\end{aligned}
\tag{7}
$$

where $V$ is a vocabulary of all possible word forms, and $\mathrm{Coh}(p_1, \ldots, p_t)$ denotes the cohort of a phoneme sequence $p_1, \ldots, p_t$ — i.e., all the words which share the given prefix of phonemes.

This model thus effectively renormalizes a language model's word-level prior $P(w \mid C)$ among words which are exactly phonologically compatible with an observed prefix. See Heilbron et al. (2022) for further details on the model specification.

## C  Inferred neural response under the prior-variable model

Figure 5 shows the inferred neural response to words of different surprisal quantiles under the prior-variable model described in Section 3.1. We see an amplified negative peak in high-surprisal words, similar to that in Figure 3 for late-recognized words.

## D  Baseline estimates of the neural response to surprisal

Figure 6 shows the baseline model's estimated response to a word's surprisal. The model recovers the standard broad negative response centered around 400 ms post word onset, which aligns with recent EEG studies of naturalistic language comprehension in both listening (Heilbron et al., 2022; Gillis et al., 2021; Donhauser and Baillet, 2020) and reading (Frank et al., 2015).

Figure 7 shows estimates of the neural response to phoneme surprisal from both the baseline model and the optimal variable model. All models tested in this paper included this phoneme surprisal predictor; the main results of the paper thus target neural activity above and beyond what is explained by phoneme-level responses. See Section 4.2 for further discussion.

## E  Choice of band-pass filter

A critical preprocessing step in our data analysis is to band-pass filter the raw EEG signal, retaining signals within a frequency window of 0.5–8 Hz. This choice of filter parameters is similar to that of other recent studies of naturalistic language comprehension which use temporal receptive field models (see e.g. Gillis et al., 2021; Heilbron et al., 2022). A reviewer points out, however, that this filter window is substantially narrower than that of classic controlled studies of the evoked N400 based on trial-averaging ERP analyses (e.g. Kutas and Hillyard, 1984; Brown and Hagoort, 1993; Brothers et al., 2023). This choice of narrow filter parameters for our temporal receptive field analysis has several motivations:

1. We wish to focus on evoked responses time-locked to events (e.g. onsets of words and

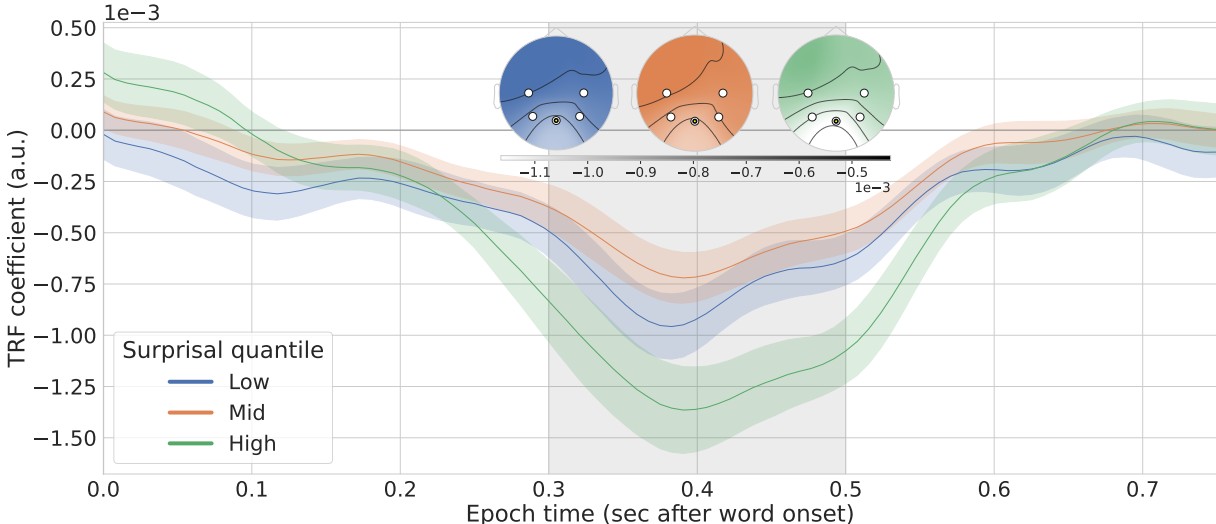

Figure 5: Modulation of scalp voltage at a centro-parietal site by surprisal for words with low ($<$ 1.33 bits, blue), mid ($<$ 3.71 bits, orange), or high ($>$ 3.71 bits, green) surprisals. Lines denote inferred coefficients of word surprisal in averaged over subjects for the sensor highlighted in the inset. Error regions denote s.e.m. ($n = 19$). Inset: spatial distribution of surprisal modulations averaged for each surprisal quantile within vertical gray regions, where less saturated colors denote more negative response. This is a replication of Figure 3 with the parameters of the prior-variable model.

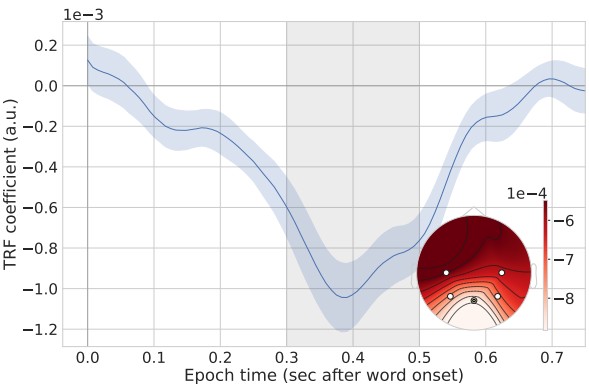

Figure 6: Modulation of scalp voltage by word surprisal in the baseline model at a central posterior sensor, highlighted in inset figure. Error regions denote s.e.m. ($n = 19$). Inset: spatial distribution of surprisal modulations averaged within vertical gray region, where less saturated colors denote more negative response.

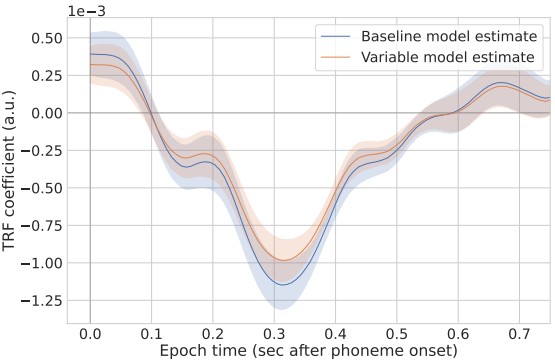

Figure 7: Modulation of scalp voltage at the same central parietal sensor used in Figure 3 by phoneme surprisal, estimated in the baseline model and the optimal variable model. Error regions denote s.e.m. ($n = 19$).

phonemes, and changes in cognitive state due to those stimuli) with rates around this frequency range. Including a wider spectrum adds variance to the signal which we cannot explain using our features of interest,

2. A high low-cut (more aggressive high-pass filter) allows us to account for signal drift; while this is handled through baselining and detrending in classic ERP analyses, temporal receptive field models have no equivalent ca-

pacity to explain drift in the signal.

However, it is possible that this choice of filter parameters could introduce artifacts in the filtered signal which affect the outcomes of our N400-focused analysis. In particular, Tanner et al. (2015) point out that aggressive high-pass filters ($\sim$ 0.5 Hz and above) can conflate evoked N400 responses with later ERPs such as the P600, and yield inflated estimates of N400 amplitude.

## E.1 Stability of the baseline model

We thus conducted a post-hoc stability analysis to better understand the sensitivity of this paradigm

| Low cut | High cut | Result |
|---------|----------|--------|
| 0.5 Hz | 8 Hz | $t = 4.91, p = 0.000113$ |
| 0.3 Hz | 8 Hz | $t = 3.26, p = 0.00435$ |
| 0.1 Hz | 20 Hz | $t = 1.95, p = 0.0666$ |
| 0.1 Hz | 8 Hz | $t = 1.84, p = 0.0826$ |

Table 3: Post-hoc stability checks on the baseline model comparison with respect to the low- and high-cut of the band pass filter.

to our choice of band-pass filter parameters. We first repeated our initial model comparison on EEG data preprocessed with different band-pass filter parameters. This model comparison evaluates the improvement in predictive performance of a temporal receptive field model which incorporates control acoustic-phonetic features and word-level features (word surprisal and frequency) above a model which does not include these word-level features. (This is the same model comparison described in the beginning of Section 3.) Table 3 shows the results of this evaluation.

We find that the predictive power of these word-level features diminishes as we decrease the low-cut frequency: beneath 0.3 Hz, this model comparison no longer shows a significant improvement in prediction due to word-level features. We do not take this result to invalidate the claim that word surprisal yields an evoked EEG response in naturalistic comprehension, since this has been supported in other studies of naturalistic comprehension with classic trial-averaging methods (Frank et al., 2015).

However, it is important to check whether the central finding of this paper — which rests on an inflated N400 amplitude in response to some types of words — is sensitive to these parameter changes. In the next section, we reproduce our main qualitative findings for those preprocessing parameters which yield a clear positive baseline outcome of the evoked N400 response to surprisal.

### E.2 Stability of our main findings

The argument of Tanner et al. (2015) would predict that the inflated N400 amplitude we observe in response to late-recognized words could be explained away as an artifact of the high-pass filter, which could confound the N400 with a later evoked response (such as the P600). If this finding were purely artifactual, then if we were to relax this high-pass filter, we should see an attenuation

of the inflated N400 response and an amplification of a P600 response.

We thus re-fit the temporal receptive field parameters of the optimal variable model described in this paper on EEG data preprocessed with a low-cut of 0.3 Hz, the lowest frequency cut at which the baseline model clearly establishes that an evoked surprisal response is readable in the signal. Figure 8 shows the estimated neural modulation by word surprisal in these preprocessed data.

We found that this variable model displayed the same qualitative patterns in neural parameters. Quantitatively, we found a similar effect size of inflated N400 amplitude (fig. 8 green line peak minus blue line peak in the shaded region; within-subject paired $t = -5.03, p = 8.71 \times 10^{-5}$).[8]

These supplementary analyses suggest that our main findings are stable to different parameterizations of a high-pass filter in EEG preprocessing.

## F Reproducibility information

We jointly estimated the parameters of the cognitive model together with the hyperparameters and parameters of the neural linking model using multivariate tree-structured Parzen estimator random search (Bergstra et al., 2011) with Optuna (Akiba et al., 2019). For subjects $i = 1, \ldots, N$, sensors $s = 1, \ldots, S$, and held-out EEG time series data for subject $i$ at sensor $s$ $Y_{i,s}$, we maximized the value $V$:

$$V = \frac{1}{N} \sum_{i=1}^{N} \left( \max_{s \in \{1,\ldots,S\}} r(Y_{i,s}, \hat{Y}_{i,s}) \right) \quad (8)$$

which is the average across subjects of the maximal Pearson correlation of predicted and observed EEG response among all sensors. Table 4 shows the precise bounds for each parameter and hyperparameter in this search procedure. We evaluated 20 trials (random settings of parameters) for the baseline model (which only incorporated the L2 coefficient), and 500 trials for all other models. The model results presented in this paper (in visualizations and statistical tests) correspond to the highest-performing outcome of each grid search.

Table 5 shows the total count of free parameters under optimization. These counts do not include the parameters of the language model used to compute word surprisal, or the word recognition model

---

[8]Our latency foundings also held null (green line peak time minus blue line peak time; within-subject paired $t = 2.17, p = 0.043$).

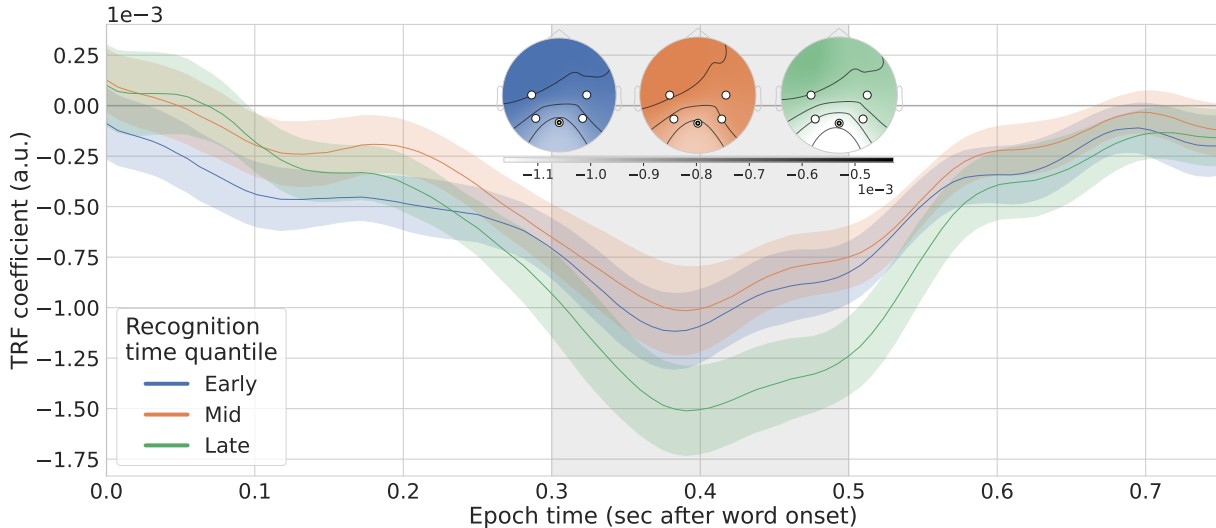

Figure 8: Modulation of scalp voltage at a centro-parietal site by surprisal for words of different recognition time quantiles (according to the variable model of Figure 3), estimated on EEG data band-pass filtered with a low-cut of 0.3 Hz.

| (Hyper)Parameter | Bounds | Notes |
|---|---|---|
| Regression L2 coefficient | $[10^2, 10^7]$ | Logarithmic space. Bounds manually selected and restricted based on early runs of each model in order to reduce total runtime |
| $\gamma$ (recognition threshold) | $(0, 1)$ | |
| $\lambda$ (evidence temperature) | $[0.1, 3]$ | |
| $\alpha$ (scatter point) | $[0, 1]$ | |
| $\alpha_p$ (prior scatter point) | $[0, 1]$ | |

Table 4: Specifications for parameter and hyperparameter bounds in random search. For details on the meaning of these parameters, see Table 1.

likelihood parameters, since these were kept fixed during optimization.

All temporal receptive field models were fit with a receptive field ranging from 0 ms to 750 ms post word onset.

We implemented all training and inference with GPU operations in PyTorch. Due to the large memory requirements of the EEG time series data and the lagged regression computations, we deployed each model fit on two NVIDIA A100 GPUs. Each of the model fits completed in two days or fewer.

| Model class | Parameter count | Decomposition |
| --- | --- | --- |
| Baseline | 138,226 | 138,225 TRF parameters + 1 hyperparameter |
| Shift | 147,446 | 147,440 TRF parameters + 5 cognitive parameters + 1 hyperparameter |
| Variable | 230,381 | 230,375 TRF parameters + 5 cognitive parameters + 1 hyperparameter |
| Prior-variable | 230,375 | 230,374 TRF parameters + 1 hyperparameter |

Table 5: Number of free parameters in all fitted models.