# OpenReview forum: "The neural dynamics of word recognition and integration"
_EMNLP/2023/Conference — EMNLP 2023 Main_

### Official Review · Reviewer_twWy · 2023-08-03

**Typos Grammar Style And Presentation Improvements:** The paper can be accepted as it is.
**Soundness:** 4

**Excitement:**

4: Strong: This paper deepens the understanding of some phenomenon or lowers the barriers to an existing research direction.

**Missing References:**

The reference list is not long, however it does contain all the most relevant contributions accrued in a couple of decades on the human processing of speech input.

**Paper Topic And Main Contributions:**

The paper proposes a computational model of auditory word processing based on a Bayesian decision model, which estimates EEG data, i.e. Event-Related Potentials representing the human brain neural activation responses during speech input.
Interestingly, the approach well replicates EEG collected data, and represents an “in-vitro” way to disentangle some still open questions in the human serial recognition process of auditory speech and parallel interpretation process.

**Questions For The Authors:**

Simply looking forward to getting access to data and script after acceptance

**Reasons To Accept:**

The paper is well structured, well written and it contributes to
an important research focus.
Results and discussion on results make it a valuable contribution.

**Reasons To Reject:**

No reason at all.

**Reproducibility:**

2: Would be hard pressed to reproduce the results. The contribution depends on data that are simply not available outside the author's institution or consortium; not enough details are provided.

**Reviewer Confidence:**

4: Quite sure. I tried to check the important points carefully. It's unlikely, though conceivable, that I missed something that should affect my ratings.

---

> ### Author Rebuttal · Authors · 2023-08-28
>
> Thank you for your helpful review! We are excited to share our analysis pipeline and invite others to remix and build on the novel modeling work we have done here.
>
> Since this review setup lacks a general commentary, we are adding this to each rebuttal: We would like to notify the reviewers of a new result. Upon further inspection of the prior-variable model (detailed in section 3.1) we discovered evaluation settings in which this model also significantly exceeds the generalization performance of the baseline model, and has a performance which does not differ significantly from that of the Variable model highlighted in the paper. (The significance tests behind these model comparisons are the same as detailed in the paper). This means that our results are no longer decisively in favor of the Variable model, and new data or new designs will be necessary in order to separate the two models that exceed baseline. In the camera ready we will present these updated comparisons and modulate our conclusions accordingly.

---

### Official Review · Reviewer_FAWT · 2023-08-04

**Soundness:** 3

**Excitement:**

3: Ambivalent: It has merits (e.g., it reports state-of-the-art results, the idea is nice), but there are key weaknesses (e.g., it describes incremental work), and it can significantly benefit from another round of revision. However, I won't object to accepting it if my co-reviewers champion it.

**Paper Topic And Main Contributions:**

This paper presents a set of model comparisons linking linguistic input to neural EEG response data recorded during natural listening / audio comprehension. The paper is well-written and the motivation for understanding the temporal and process-level dynamics of spoken-word / speech recognition are well presented.

I will note first those on a procedural note that the paper does slightly exceed the 8 page limit for the conference, with the limitations section going onto the 9th page.

On the content side, the brunt of the work is of interest primarily to a cognitive neuroscience (or neuroscience of language) audience rather than those attending EMNLP, even in the psycholinguistics track. I found a some of the neuro-details to be outside my area of expertise, for instance the details of the TRF model. While I am by no means a neuroscientist, this makes me wonder how approachable the topic will be to the general audience at the conference. Furthermore, my comments about the paper should be taken with a grain of salt (given my lack of neuroscience data modeling experience).

That said I'm a little stuck about how to situation or interpret this work. It seems to me to be a model-comparison of fits to EEG data, but that the primary contribution is a method for uncovering statistical patterns present in existing EEG data, rather than a cognitive model which explains what's going on in the listeners mind. In particular, I'm wondering if the model makes any novel, testable predictions? Or if the fit of the model to the EEG data is being assessed purely on the basis of consistency. How much of the model's performance is parameter-dependent, and which assumptions made at the model architecture level -- i.e. framing the task in terms of Bayesian decision theory -- are necessary for explanation or are something that while formally elegant are incidental to the output.

**Reasons To Accept:**

The paper is well-written and, as far as I am able to evaluate, the methods are sound. This modeling/statistical analysis sheds light on questions about the neural and temporal dynamics of spoken word recognition.

**Reasons To Reject:**

I'm not sure if this fits well with the primary topics of the conference and as I note above I'm a little stuck about how to situation or interpret this work. It seems to me to be a model-comparison of fits to EEG data, but that the primary contribution is a method for uncovering statistical patterns present in existing EEG data, rather than a cognitive model which explains what's going on in the listeners mind. In particular, I'm wondering if the model makes any novel, testable predictions? Or if the fit of the model to the EEG data is being assessed purely on the basis of consistency. How much of the model's performance is parameter-dependent, and which assumptions made at the model architecture level -- i.e. framing the task in terms of Bayesian decision theory -- are necessary for explanation or are something that while formally elegant are incidental to the output.

**Reproducibility:**

3: Could reproduce the results with some difficulty. The settings of parameters are underspecified or subjectively determined; the training/evaluation data are not widely available.

**Reviewer Confidence:**

3: Pretty sure, but there's a chance I missed something. Although I have a good feel for this area in general, I did not carefully check the paper's details, e.g., the math, experimental design, or novelty.

---

> ### Author Rebuttal · Authors · 2023-08-28
>
> Thank you for your helpful review.
>
> Regarding audience: our work is part of a growing trend combining neuroscientific data with modern NLP models for psycholinguistic insights [1,2,3,4,6,7]. We believe these methods and data sources can be useful for addressing core architectural questions in language processing, and that our paper helps demonstrate this utility. We will work to clarify and extend the neural model presentation in the final version to be more friendly to readers unfamiliar with neural data.
>
> Regarding predictions and interpretation: one half of our multi-level model is stated at the cognitive level, describing the perceptual state of the listener at any point in stimulus-time. It makes critical predictions about the specific point in each word at which a listener can successfully identify it. These predictions can be compared with other empirical measures of word recognition (e.g. [5]), which we plan to address in future work. For the moment, we see these cognitive-level predictions pay off in the way they help us better explain EEG neural responses to words, under the Variable linking model. This success is indirect evidence for the reality of these cognitive predictions.
>
> Regarding what is essential to the model: it's true that this model is one of many possible cognitive models. We chose this model for its simple and elegant computational-level explanation of the recognition phenomenon. Future work should more thoroughly explore the space of possible models; as this paper stands, we believe that our combination of a structured cognitive model with high-dimensional neural analysis is already an important methodological contribution.
>
> [1] Gauthier, J., & Levy, R. (2019). Linking artificial and human neural representations of language. Conference on Empirical Methods in Natural Language Processing.
> [2] Toneva, M., & Wehbe, L. (2019). Interpreting and improving natural-language processing (in machines) with natural language-processing (in the brain). NeurIPS.
> [3] Jain, S., & Huth, A.G. (2018). Incorporating Context into Language Encoding Models for fMRI. bioRxiv.
> [4] LeBel, A., Jain, S., & Huth, A.G. (2021). Voxelwise Encoding Models Show That Cerebellar Language Representations Are Highly Conceptual. The Journal of Neuroscience, 41, 10341 - 10355.
> [5] Tucker, B.V., Brenner, D., Danielson, D.K., Kelley, M.C., Nenadić, F., & Sims, M. (2018). The Massive Auditory Lexical Decision (MALD) database. Behavior Research Methods, 51, 1187-1204.
> [6] Wehbe, L., Murphy, B., Talukdar, P.P., Fyshe, A., Ramdas, A., & Mitchell, T.M. (2014). Simultaneously Uncovering the Patterns of Brain Regions Involved in Different Story Reading Subprocesses. PLoS ONE, 9.
> [7] Fyshe, A., Talukdar, P.P., Murphy, B., & Mitchell, T.M. (2014). Interpretable Semantic Vectors from a Joint Model of Brain- and Text- Based Meaning. Proceedings of the conference. Association for Computational Linguistics. Meeting, 2014, 489-499 .
>
> Since this review setup lacks a general commentary, we are adding this to each rebuttal: We would like to notify the reviewers of a new result. Upon further inspection of the prior-variable model (detailed in section 3.1) we discovered evaluation settings in which this model also significantly exceeds the generalization performance of the baseline model, and has a performance which does not differ significantly from that of the Variable model highlighted in the paper. (The significance tests behind these model comparisons are the same as detailed in the paper). This means that our results are no longer decisively in favor of the Variable model, and new data or new designs will be necessary in order to separate the two models that exceed baseline. In the camera ready we will present these updated comparisons and modulate our conclusions accordingly.

---

### Official Review · Reviewer_oXeZ · 2023-08-06

**Soundness:** 4

**Excitement:**

4: Strong: This paper deepens the understanding of some phenomenon or lowers the barriers to an existing research direction.

**Paper Topic And Main Contributions:**

This paper presents a bayesian model of context-sensitive word recognition and evaluates variants of this model which differ in how word-recognition and contextual factors modulate ERP signals such as the N400. Overall this is an exciting integration of psycholinguistic theories of word recognition, computational modeling, and highly time-resolved neural signals. The results suggest that word recognition and contextual integration may begin even before word-recognition processes are completed.

**Questions For The Authors:**

Given the robust existing modeling work on single-word recognition, I would have appreciated more explicit discussion of what is original here about the non-contextual elements of the model, and what is shared with existing frameworks. (On my understanding, the current proposal is akin to a continuous feed-forward architecture along the lines of RACE, but lacks feedback connections a-la TRACE and differs from COHORT in that word recognition is not constrained to begin after the first syllable). I'm not sure if my understanding is correct, but it seems like there is a missed opportunity to engage with some of these debates.


**Reasons To Accept:**

- Tight connection between theoretical foundation, modeling work, and neural signals
- Model comparison provides clear theoretical deliverable in terms the relationship between word recognition and contextual integration


**Reasons To Reject:**

The EEG data are highly band-limited to 0.5-8 Hz. This is an emerging norm in TRF analyses such as those presented here, but different substantially from the band used for "standard" ERP analysis (e.g. 0.1 - 20 Hz); Tanner et al. 2015 Psychophysiology in particular point out that both phase and amplitude distortions on the evoked response that can be caused by such a high low-cut. I for one would appreciate a robustness analysis to better understand how sensitive the model fitting results are to the choice of filter here.

**Reproducibility:**

3: Could reproduce the results with some difficulty. The settings of parameters are underspecified or subjectively determined; the training/evaluation data are not widely available.

**Reviewer Confidence:**

3: Pretty sure, but there's a chance I missed something. Although I have a good feel for this area in general, I did not carefully check the paper's details, e.g., the math, experimental design, or novelty.

---

> ### Author Rebuttal · Authors · 2023-08-28
>
> Thank you for your helpful review.
>
> Regarding the band pass filter: this is an important point. We are not able to perform this robustness analysis (which entails re-running our entire pipeline) within this short rebuttal period, but will do this prior to the camera ready and report these results in the paper, preferentially over the current methods with a high low-cut.
>
> Regarding the space of recognition models: you are broadly correct. This is a feed forward race architecture operating on the level of a single word, akin to Shortlist B [1]. It is explicitly Bayesian, unlike TRACE, and is similar to a cohort model, except the likelihood function allows for substantial mismatches between acoustic input and a perceived word. We will add this detail and references in the final paper. We are excited to more thoroughly explore and compare the space of computational recognition models (in predicting neural data!) in future work.
>
> Since this review setup lacks a general commentary, we are adding this to each rebuttal: We would like to notify the reviewers of a new result. Upon further inspection of the prior-variable model (detailed in section 3.1) we discovered evaluation settings in which this model also significantly exceeds the generalization performance of the baseline model, and has a performance which does not differ significantly from that of the Variable model highlighted in the paper. (The significance tests behind these model comparisons are the same as detailed in the paper). This means that our results are no longer decisively in favor of the Variable model, and new data or new designs will be necessary in order to separate the two models that exceed baseline. In the camera ready we will present these updated comparisons and modulate our conclusions accordingly.
>
> [1] Norris, D., & McQueen, J.M. (2008). Shortlist B: a Bayesian model of continuous speech recognition. Psychological review, 115 2, 357-95 .

---

### Meta-Review · Area_Chair_Do7s · 2023-09-18

**Recommendation:** 5

**Metareview:**

> Overview

This paper presents a cognitive model of word integration in context, which makes predictions about word recognition times. It then proposes two linking functions (termed *shift model* and *variable model*) of how their predicted word recognition times affect observed neural responses, which they compare to a baseline (which does not use word recognition time information). They find that the variable model significantly outperforms the baseline.

> Meta Review

The three reviewers and I agree that this paper’s analysis sheds light on the neural and temporal dynamics of spoken word recognition. They also agree that the paper’s methods are sound (modulo a robustness analysis the authors promised to address for camera ready). There is some disagreement, however, regarding how to interpret the paper’s theoretical contributions. While two of the reviewers appreciate the proposed cognitive model for word recognition in context, reviewer FAWT questions which kind of testable predictions it makes. Personally, I share some of reviewer FAWT’s scepticism about the proposed model (longer comment below). Regardless of these issues, however, I found this paper interesting and believe it might motivate more future work on the topic.

> Further Comments


I personally believe that: (i) this paper's topic does fit EMNLP; (ii) this paper's analysis makes a significant contribution in investigating the temporal dynamics of spoken word recognition; and (iii) this paper's contributions are not immediately clear from its text.

Regarding (iii), I had questions similar to reviewer FAWT's, not being able to clearly situate this paper's contributions. At a first read of this paper, I believed that this paper’s main contribution was the two proposed linking functions (described in section 1.2) and the comparison of model fits to EEG data. After re-reading all reviews and author responses in detail, however, I now believe that the paper also claims its Bayesian cognitive model (in section 1.1) as a key contribution. If this is the case, I would expect (as noted by reviewer oXeZ) a longer discussion comparing this model with prior work modelling single-word and word-in-context recognition. Ideally, I would expect to even see an empirical comparison between how predictive of neural responses are the *word recognition times* output by this cognitive model vs prior works’. The proposed cognitive model (in particular, eqs. 1 and 2 here) also seems very similar to, e.g., the word recognition model from Smith and Levy (2010)—which was developed for analysing reading times—and it would be nice to see a discussion comparing them. Further, the *variable model* could be more carefully motivated in my opinion. In particular, I don't see why the word integration process would rely on clusters based on a word's required recognition time, instead of, e.g., varying smoothly as a function of the word's recognition time.

Finally, I believe that reviewer FAWT’s question about how much of the model's performance is parameter-dependent becomes particularly relevant given the new results (in the authors’ response) that the proposed *variable model* (which predicts neural signals using clusters of word recognition times) is not significantly better than the *prior-variable model* (which predicts neural signals using clusters of surprisal values).

Smith and Levy (2010). Fixation durations in first-pass reading reflect uncertainty about word identity.

> Potential Typos

I believe there might be a typo in eq. 5, as I don't see how word-level features would be incorporated into this equation.

---

### Decision · Program_Chairs · 2023-10-07

**Decision:**

Accept-Main

**Comment:**

> Overview

This paper presents a cognitive model of word integration in context, which makes predictions about word recognition times. It then proposes two linking functions (termed *shift model* and *variable model*) of how their predicted word recognition times affect observed neural responses, which they compare to a baseline (which does not use word recognition time information). They find that the variable model significantly outperforms the baseline.

> Meta Review

The three reviewers and I agree that this paper’s analysis sheds light on the neural and temporal dynamics of spoken word recognition. They also agree that the paper’s methods are sound (modulo a robustness analysis the authors promised to address for camera ready). There is some disagreement, however, regarding how to interpret the paper’s theoretical contributions. While two of the reviewers appreciate the proposed cognitive model for word recognition in context, reviewer FAWT questions which kind of testable predictions it makes. Personally, I share some of reviewer FAWT’s scepticism about the proposed model (longer comment below). Regardless of these issues, however, I found this paper interesting and believe it might motivate more future work on the topic.

> Further Comments


I personally believe that: (i) this paper's topic does fit EMNLP; (ii) this paper's analysis makes a significant contribution in investigating the temporal dynamics of spoken word recognition; and (iii) this paper's contributions are not immediately clear from its text.

Regarding (iii), I had questions similar to reviewer FAWT's, not being able to clearly situate this paper's contributions. At a first read of this paper, I believed that this paper’s main contribution was the two proposed linking functions (described in section 1.2) and the comparison of model fits to EEG data. After re-reading all reviews and author responses in detail, however, I now believe that the paper also claims its Bayesian cognitive model (in section 1.1) as a key contribution. If this is the case, I would expect (as noted by reviewer oXeZ) a longer discussion comparing this model with prior work modelling single-word and word-in-context recognition. Ideally, I would expect to even see an empirical comparison between how predictive of neural responses are the *word recognition times* output by this cognitive model vs prior works’. The proposed cognitive model (in particular, eqs. 1 and 2 here) also seems very similar to, e.g., the word recognition model from Smith and Levy (2010)—which was developed for analysing reading times—and it would be nice to see a discussion comparing them. Further, the *variable model* could be more carefully motivated in my opinion. In particular, I don't see why the word integration process would rely on clusters based on a word's required recognition time, instead of, e.g., varying smoothly as a function of the word's recognition time.

Finally, I believe that reviewer FAWT’s question about how much of the model's performance is parameter-dependent becomes particularly relevant given the new results (in the authors’ response) that the proposed *variable model* (which predicts neural signals using clusters of word recognition times) is not significantly better than the *prior-variable model* (which predicts neural signals using clusters of surprisal values).

Smith and Levy (2010). Fixation durations in first-pass reading reflect uncertainty about word identity.

> Potential Typos

I believe there might be a typo in eq. 5, as I don't see how word-level features would be incorporated into this equation.